# High Prevalence and Genetic Variability of *Hepatozoon canis* in Grey Wolf (*Canis lupus* L. 1758) Population in Serbia

**DOI:** 10.3390/ani12233335

**Published:** 2022-11-29

**Authors:** Milica Kuručki, Snežana Tomanović, Ratko Sukara, Duško Ćirović

**Affiliations:** 1Faculty of Biology, University of Belgrade, 11000 Belgrade, Serbia; 2Group for Medical Entomology, Centre of Excellence for Food- and Vector-Borne Zoonoses, National Institute of Republic of Serbia, Institute for Medical Research, University of Belgrade, 11129 Belgrade, Serbia

**Keywords:** grey wolf, *Hepatozoon*, Serbia, *Canis lupus*

## Abstract

**Simple Summary:**

There are a number of different animal species that can be infected by *Hepatozoon canis*, but the most common are domestic and wild carnivores. This study presents the first results of the occurrence, distribution, and diversity of *H. canis* in Serbia and contributes to the limited knowledge regarding the role of grey wolves in the enzootic cycles of this pathogen. Based on a comparison with previously published data, we found a higher prevalence and higher genetic diversity of *H. canis*. The obtained results showed an overall high prevalence of the pathogen, with 57.94% of tested animals positive for *H. canis*, while genetic analysis of a sequenced fragment of the 18S ssrRNA gene showed variability at five positions leading to five sequence types present in grey wolves, where only two were previously known of. In addition to the known patterns of transmission of this pathogen, through tick ingestion during grooming or the transplacental route, the high diversity of *H. canis* in Serbia could be explained by the diet of grey wolves in this area. Further studies are needed to determine the mechanism of transmission, the potential source of infection, and the impact of this pathogen on wild carnivores.

**Abstract:**

Wild canids are globally recognised as hosts and reservoirs of a large number of ecto- and endoparasites. Data that reveal the importance of the grey wolf (*Canis lupus* L.1758) in the spread of hepatozoonosis are very scarce. There are a large number of different potential host species that can be infected by *Hepatozoon canis*, but the most common are domestic and wild carnivores, such as dogs, jackals, foxes, and wolves. In this study, the epidemiological significance of the grey wolf as a host for the pathogen was analysed for the first time in Serbia, as well as the genetic variability of *H. canis*. The presence of *H. canis* in wolf spleens has been demonstrated using molecular methods. A total of 107 wolf spleen samples from 30 localities in Serbia were analysed. The presence of *H. canis* was confirmed in 62 (57.94%) individuals from 26 out of 30 localities. According to the analysis, the sampled *H. canis* sequences were found to be characterised by a certain heterogeneity. Based on five mutated nucleotide sites in the sequences, *H. canis* could be divided into five sequence types, S1 to S5. The five sequence types can potentially circulate in grey wolf populations as well as among other domestic and wild canids. This study is the first confirmation of the presence of *H. canis* in grey wolf populations in Serbia. Considering that the role of this vector-borne disease is poorly researched in wild carnivores, it is very important to indicate the role of this species in the circulation of this pathogen in natural ecosystems.

## 1. Introduction

Species of the genus *Hepatozoon* belong to the Apicomplexa phylum, a large and diverse group of unicellular sporulating parasitic protists that infect different animal species. So far, more than 300 species of the genus *Hepatozoon* have been described [1]. This group of parasites primarily infects mammalian leukocytes and amphibian, reptilian, and avian erythrocytes [2]. Hepatozoonosis, the disease caused by *Hepatozoon* parasites, is prevalent in dogs and affects cats to a much lesser degree. Among the wild canids, *Hepatozoon* spp. has been detected in grey wolf (*Canis lupus*) [3], golden jackal (*Canis aureus* L. 1758) [4], black-backed jackal (*Lupulella mesomelas* S. 1775) [5], bush dog (*Speothos venaticus* L. 1842), maned wolf (*Chrysocyon brachyurus* I. 1815) [6] and the red fox (*Vulpes vulpes* L. 1758) [7]. Hepatozoonosis in dogs is caused by the widespread species *Hepatozoon canis* J. 1905 in temperate and tropical regions [8] and by *Hepatozoon americanum* V.J. 1997 in North America [9]. Clinical presentation varies from asymptomatic infection in seemingly healthy animals to life-threatening disease, but *H. canis* in dogs mainly presents a mild clinical picture. The disease affects the spleen, lymph nodes, and bone marrow, which, in severe cases, leads to anaemia and lethargy, anorexia, fever, lymphadenomegaly, and weight loss [10,11,12]. Ticks, mites, sand flies, tsetse flies, mosquitoes, fleas, lice, reduviid bugs, and leeches are definitive hosts of species of the genus *Hepatozoon* [1]. The most common vector of the protozoan *H. canis* is the brown dog’s tick *Rhipicephalus sanguineus* L. 1806 [2,13], and pathogen occurrence most often coincides with the geographic distribution of tick hosts [14]. In addition, transplacental transmission of pathogens from mother to offspring has been shown [15]. Unlike in other tick-borne diseases, where transmission from vector to the host is mostly hematophagous, the main route of transmission of *H. canis* from ticks as a vector to the vertebrate host is oral, by ingestion of an infected tick in whose body cavity protozoic oocysts are found [16]. It is assumed that the animal gets infected by ticks while grooming its coat or feeding on infected prey [13]. Transmission of the pathogen from hosts to ticks occurs by the hematophagous route during feeding on the infected animal.

Recent research has shown a high prevalence of *H. canis* in red foxes in Serbia, while no data are available for the other canids that are present [17]. Together with the golden jackal and red fox, the grey wolf is one of the three autochthonous species of wild carnivores in Serbia.

The grey wolf is a widely distributed top predator that has historically been distributed over almost the whole of Eurasia and North America [18]. Its distribution has since been reduced due to the widespread campaign of eradication by poisoning and killing in the past, habitat loss and fragmentation and the decline in natural prey populations. In Europe, the most abundant population is preserved in Eastern and Southern Europe, while in Western Europe, only a small and isolated population has survived [18]. Due to conservation efforts, legal protection, and supportive public opinion, the European grey wolf population has recovered in the last few decades [19], and wolves have recolonised several areas where they had earlier been exterminated [20,21].

The distribution range of the grey wolf in Serbia is relatively continuous and includes forested hilly and mountainous areas in the eastern, southern, and western parts of the country, as well as a small, isolated population in the southeastern Banat region. According to recent estimates, the wolf population has tended to be stable or is even increasing slightly, with a population of approximately 800–900 individuals.

Information about *H. canis* in grey wolf populations is very limited, with only a single study published up to now [3]. Hence, with our study, we aim to contribute to improving understanding of the role of the grey wolf as a host of *H. canis*, the importance of this canid species in the occurrence and spread of hepatozoonosis (both within its species and to other canids), and to characterise the prevalence of this pathogen in the grey wolf population in Serbia, both in terms of its spatial presence as well as genetic variability.

## 2. Material and Methods 

### 2.1. Study Area and Sample Preparation

Over a period of 10 years (2010–2019), spleen samples were collected from legally shot grey wolves in collaboration with local hunters. To avoid degradation, a complete spleen was collected shortly after shooting. The organs were kept adequately in marked bags with all the necessary information (date of death, sex, location) about shot animals. The samples were transferred in a cold chain to the laboratory of the Faculty of Biology, University of Belgrade, and stored at −20 °C prior to further analysis. Up to 10 µg of frozen spleen from individual animals was homogenised using sterile pestles and subjected to DNA extraction using a Gene Jet Genomic DNA Purification Kit (Fermentas, Thermo Fisher Scientific, Waltham, MA, USA) according to the manufacturer’s instructions. *H. canis* DNA was detected using conventional PCR with primers set that amplify the 666-bp fragment of the 18S ssrRNA gene, HepF_for (5′-ATACATGAGCAAAATCTCAAC-3′) and HepR_rev (5′-CTTATTATTCCATGCTGCTGCAG-3′) [22]. The amplification reaction mixture was amplified and consisted of 24.75 μL of nuclease-free water, 10 μL of 5 X Green Reaction Buffer (7.5 mM MgCl_2_; Ph 8.5), 1 μL of dNTP’s (10 mM), 0.250 μL of Taq. polymerase (5u/μL, GoTaq G2 DNA Polymerase, Promega Corporation, Madison, WI, USA), 4 μL HepF_for primer (10 pmol/μL), 4 μL HepR_rev primer (10 pmol/μL) and 6 μL template DNA. The amplification conditions were as follows: initial denaturation at 95 °C for 2 min, then 40 cycles of denaturation at 95 °C for 1 min, annealing at 56 °C for 1 min, elongation at 72 °C for 1 min and final elongation at 72 °C for 5 min. Amplification was performed in an Eppendorf 5333 MasterCycler Thermal Cycler (Eppendorf, Hamburg, Germany). Amplified products were visualised on 2% agarose gels.

### 2.2. Sequencing and Sequence Processing

Samples positive for *H. canis* DNA were sent for sequencing to the Macrogen Commercial Laboratory (Amsterdam, The Netherlands). Further processing of the sequences was conducted in FinchTV software (version 1.5.0, Geospiza Inc., Seattle, WA, USA). Sequences of grey wolves from Serbia were compared with available sequences from the GenBank using the BLAST search (National Center for Biotechnology Information, http://www.ncbi.nlm.nih.gov/BLAST, accessed on 1 October 2022) [23]. Phylogenetic analyses and construction of Neighbor-Joining (1000 bootstrap replicates) based on the Tamura 3- parameter (T92) model were obtained using MEGA X software (Pennsylvania State University, State College, PA, USA) [24]. All representative sequences from this study are deposited in GenBank and are available under the following accession numbers (OP012773, OP012774, OP012775, OP012776, OP012777, OP012778, OP012779, OP012780, OP012781, OP012782, OP012783, OP012784, OP012785, OP012786, OP012787, OP012788, OP012789, OP012790, OP012791, OP012792, OP012793, OP012794, OP012795, OP012796, OP012797, OP012798, OP012799, OP012800, OP012801, OP012802).

### 2.3. Statistical Analysis

For statistical analyses, and considering that the wolf population in Serbia is divided into two groups by the natural border of the Great Morava river and South Morava river, samples were divided into two subpopulations—eastern and western (Figure 1). Chi-square tests were used to assess differences between registered prevalences in these two subpopulations in addition to the prevalence in male and female hosts. Data were analysed using Statistica 5.1 (Statsoft, Tulsa, OK, USA), with the level of significance being *p* < 0.05.

## 3. Results

Spleen samples from 107 grey wolves, 44/107 (41.12%) females and 63/107 (58.88%) males were collected for analysis. Animals originated from a wide area in Serbia and were hunted at 30 different locations. For further analysis, the area of Serbia was divided into the western and eastern parts according to the position of the Great Morava and South Morava rivers. Fourteen localities belonged to the eastern part and sixteen to the western part of the study area (Figure 1).

PCR analyses showed that 62/107 (57.94%) samples were positive for the presence of *H. canis*, originating from 24/44 (54.55%) female and 38/63 (60.32%) male animals. No statistically significant differences in the prevalence of *H. canis* infections were detected between the sexes (*p* > 0.55176). Animals infected with *H. canis* originated from 26 localities, 13 each from the eastern and western parts. However, there were statistically significant differences in the prevalence of infection of this pathogen between the eastern subpopulation, 32/44 (72.7%), and the western subpopulation, 30/63 (47.6%) (*p* > 0.00963).

The prevalence at individual localities ranged from 28.57% to 100%. The highest prevalence was recorded for 13 sites (Blace, Boljevac, Golubac, Jagodina, Kruševac, Paraćin, Petrovac na Mlavi, Pirot, Raška, Sokobanja, Svrljig, Valjevo, and Zubin Potok), where all animals tested positive (100%); however, the number of animals collected at each of these localities was rather low, from one to six. The largest number of samples was collected from the locality Sjenica, where out of a total of 30 tested animals, 11 (36.66%) were positive for *H. canis*. None of the tested samples collected at four localities (Leposavić, Aleksandrovac, Ražanj, and Gornji Milanovac) were positive for *H. canis*, though the number of animals originating from these localities was also low, one or two from each locality (Table 1).

A total of 36 representative positive samples were subjected to sequencing. The partial sequences of the 18S rRNA gene of *H. canis*, with lengths ranging from 454 to 627 bp, showed a certain heterogeneity. The alignment was built based on 30 sequences covering the length of 539 bp, where positions of sequence variability were observed, while six sequences were excluded from the analysis due to insufficient length on either the 5′ or 3′ end. The sequences showed variability at five positions, corresponding to five different sequence types, S1 to S5 (Figure 2, Table 2). Four sequences (4/30, 13.3%) belong to each of the S1, S4, and S5 sequence types, 17/30 (56.7%) belong to the S2 sequence type, and one sequence (1/30, 3.4%) belong to the S3 sequence type. Concerning regional distribution, 20 sequences originated from animals hunted at sites in the eastern region and 10 from sites in the western region. All five sequence types were present in the western region, while four (S1, S2, S4, and S5) were present in the eastern region. Although the S2 sequence type was dominant in both regions, slight differences were observed in terms of the distribution of sequence types between regions, with the domination of S3 and S4 in the west and S1, S2, and S5 in the east.

For further analysis, five representative sequences were selected for comparison, one from each sequence type, and found to share 100% identity and coverage with previously published sequences available in GenBank (Figure 3). Based on the results of BLAST searching, S1 sequences aligned with sequences from dogs from Croatia, Hungary, and Germany (FJ497010, MK301151, MK757806), and foxes from Slovakia and Germany (KX879141, MK757792); S2 with sequences from dogs from Croatia and Germany (FJ497009, MK757805), foxes from the Czech Republic and Hungary (KU893119, KF322142), a wolf from Croatia (MH656730), and *Ixodes ricinus* ticks from Slovakia and Czech Republic (MG253004, KU597242). The S3 sequence type aligned with only two sequences from GenBank, both from foxes from Italy (KP715303, KP644235). The S4 sequence type showed 100% identity to sequences from dogs from Germany, South Korea, Cuba, China, and Malaysia (MK757802, MK238384, MN393911, MZ675626, KT267961), foxes from Serbia and Slovakia (MH699891, KX887327), crab-eating fox (*Dusicyon thous*) from Brazil (AY461375), and *Rhipicephalus sanguineus* tick from a dog from Egypt (MG564217). The S5 sequence was identical to sequences from dogs from Iran, Zambia, Kyrgystan, Turkey, Hungary, and Croatia (KU360328, LC331054, MG917718, KY247117, MK301149, FJ497019), fox and golden jackal from Romania (KM096414, KX712129), a wolf from Croatia (MH656729), and *Rhipicephalus sanguineus* tick from a dog from Egypt (MG564216).

Comparison of sequences obtained in our study with sequences from wolves from Germany (MN791088, MN791089), designated as G1 and G2 in the paper of Hodžić et al. [3], was possible for the partial length covering four out of five variable positions (380 bp, 400 bp, 522 bp, and 523 bp). Excluding the variable position located at the 3 bp site from the analysis, it was not possible to distinguish sequence types S1 and S2 among sequences from Serbia, and the German G1 sequence showed 100% identity with the S1 and S2 sequences, while the G2 sequence was identical with the S4 sequence type. 

## 4. Discussion

In this study, the presence, high prevalence, as well as high genetic variability of *H. canis* in grey wolf populations were characterised for the first time in Serbia. The obtained results showed an overall high prevalence of pathogen, with as many as 62 (57.94%) individuals out of a total of 107 testing positive for *H. canis*. The presence of hepatozoonosis caused by *Hepatozoon canis* has been observed throughout Europe, Asia, and Africa [8]. Most studies in Europe have been performed on foxes, and a high prevalence was detected throughout Europe [25], while the presence of this pathogen in the golden jackal has only been confirmed in a few countries [26], and a high prevalence (46.0%) of *H. canis* was recently reported for the grey wolf in Germany [3]. Previous research has noted a relatively low degree of variability in *H. canis* strains, and it is not yet clear whether there is a correlation between certain genotypes with host species or whether they circulate between different species [27]. The presence of identical *H. canis* haplotypes in different host species indicates possible direct or horizontal transmission through these species [27,28]. The most significant assumed vector, the tick *R. sanguineus,* is characterised by cosmopolitan distribution and high adaptability to different environmental conditions. Although it most often parasitises dogs, it can infect a large number of other domestic and wild animals and sporadically parasitise humans. Recent studies based on molecular and morphological analyses, as well as cross-breeding experiments, have indicated that *R. sanguineus* is not a single species but a complex of at least two species with the existence of additional operative taxonomic units within these clusters [29]. While phylogeographic analyses have shown a clear geographical separation of “moderate” and “tropical” logs due to mean annual temperatures [30], the distribution of lower taxonomic units within the logs is influenced by other environmental factors, most likely the distribution of haplotypes of vertebrate hosts.

In our study, in addition to the high prevalence of *H. canis* in grey wolf populations, high genetic variability of *H. canis* was also observed. The identified sequences showed variability at five positions where we obtained five sequence types (S1–S5). Since previous studies have not shown great genetic variability of this pathogen, the presence of five different sequence types of *H. canis* has been proven for the first time in Serbia. The predation transmission for *Hepatozoon ayorgbor* S. 2007 was demonstrated by Sloboda et al. [31] in their study of experimental infection of snakes with the tissue of infected rodents; thus, the existence of more than one route of transmission for *H. canis* is also an option. The high genetic variability of *H. canis* in Serbia, obtained in our study, can be observed in light of the grey wolf diet. In the previous study, it was elucidated that the most common natural prey species of the grey wolf in Serbia are roe deer (*Capreolus capreoulus* L. 1758) and wild boar (*Sus scrofa* L. 1758), and less often (*Lepus europaeus* P. 1778) and small rodents [32]. These species are known as common hosts for ticks, including *R. sanguines*, the predominant vector of *H. canis* [33,34]. Further, wild boar is an omnivorous species with the domination of plants in its diet (93%), but it also consumes animal material, including reptiles, small rodents, and carcasses of game animals, which are known to harbour *Hepatozoon* spp. [35,36] and thus have the potential to contribute to the transmission chains of *H. canis*. Taking all of the above into consideration, the possibility that the grey wolf may become infected by consuming infected ticks together with prey tissue (in particular, skin) cannot be ruled out. The several trophic levels in the food chain, together with multiple transmission routes, could contribute to the high parasitic variability in the grey wolf as a top predator species, as observed in the present study.

This route of transmission raises new questions and provides space for further research and detection of this pathogen in other animal species, which could provide evidence supporting this mode of transmission.

The S4 sequence type of the *H. canis* pathogen found in the grey wolf in Serbia in this study is identical to the nucleotide sequences of *H. canis* that have been detected in the red fox population from Serbia [17]. It is not possible to pinpoint the primary source of infection in canids, but this finding indicates a possible common pattern of transmission of the pathogen *H. canis*. This is the first study on vector-borne disease hepatozoonosis in grey wolf populations in Serbia, and, to the best of our knowledge, this is the second such study in the world. *H. canis* was found to be widespread in this wild carnivore species, as was previously shown for red foxes in Serbia [17].

## 5. Conclusions

According to the obtained results, a question arises about the possible emergence of a common pattern of transmission of *H. canis* between other wild and domestic carnivores. Red foxes and golden jackals are considered the main hosts of this protozoan parasite. According to this study, the role of the grey wolf in the enzootic cycle of *H. canis* is similar to that of the other two carnivores. Further studies are needed to determine the potential source of infection and the impact of this pathogen on the conservation of wild carnivores and its mechanism of transmission within and between species.

## Figures and Tables

**Figure 1 animals-12-03335-f001:**
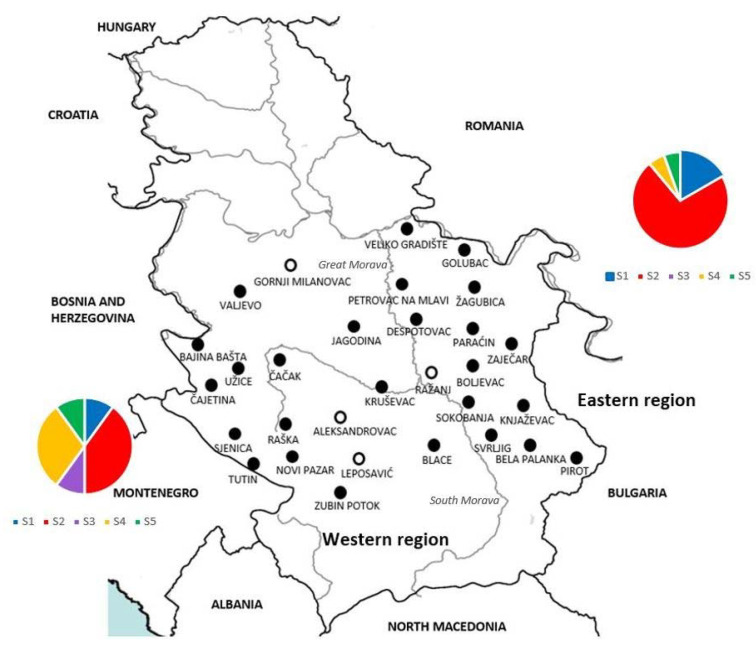
Geographical distribution and regional belonging to the western and eastern regions of localities in Serbia where grey wolves were hunted. The area of Serbia was divided into the western and eastern parts according to the position of the Great Morava and South Morava rivers. Circles—positive (
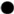
) and negative (
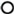
), indicate the finding of DNA of *Hepatozoon canis* in spleen samples from each locality. Charts present distribution of sequence types S1–S5 in western (**left**) and eastern (**right**) part of Serbia).

**Figure 2 animals-12-03335-f002:**
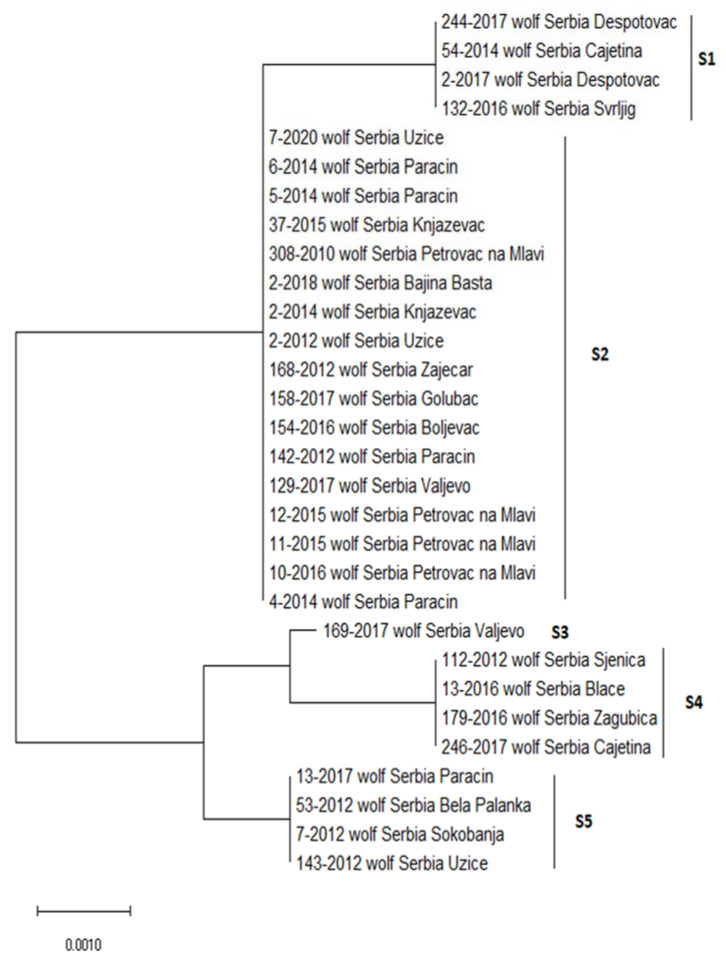
Neighbour-joining analysis of partial 18S rRNA sequences showing the five genotypes of *Hepatozoon canis* isolates found in this study. Sequence types (S1–S5) are marked by the group number on the right side of the figure. The code numbers and locality are indicated for each animal.

**Figure 3 animals-12-03335-f003:**
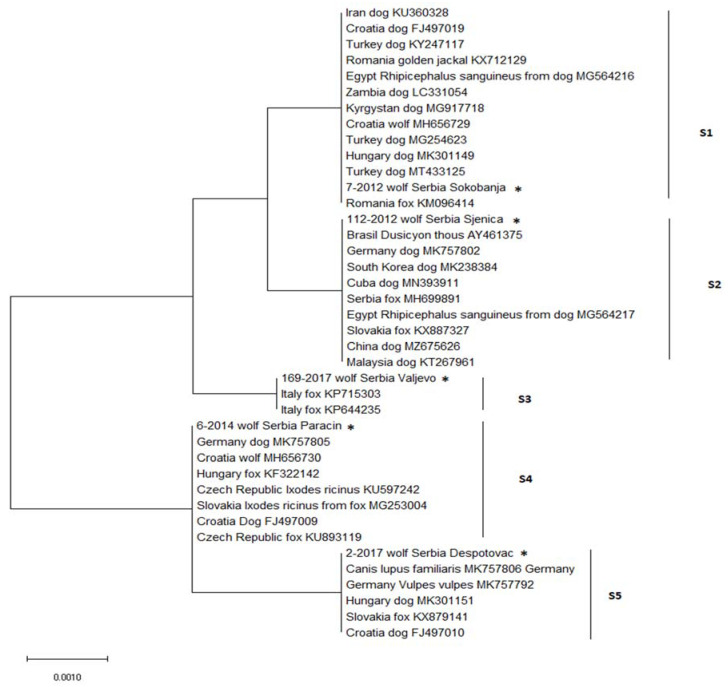
Neighbour-joining analysis of partial 18S rRNA showing the distances between *Hepatozoon canis* isolates from Serbia and other related isolates from GenBank. Host species, accession number, and country of origin are given for each nucleotide sequence from GenBank. From Serbia, one representative sequence of each sequence type marked with its code number was used and each is marked with *.

**Table 1 animals-12-03335-t001:** Distribution of grey wolf (*Canis lupus*) samples based on sampling locality, gender, and prevalence of *Hepatozoon canis*.

Region	Locality	No. of Collected Samples	M/F	No. of Positive Samples *Hepatozoon canis* (%)	M/F
WEST REGION	Bajina Bašta	2	2/0	1 (50%)	1/0
	Valjevo	2	2/0	2 (100%)	2/0
	Gornji Milanovac	1	1/0	/	0/0
	Čajetina	7	5/2	4 (57.14%)	3/1
	Užice	6	5/1	3 (50%)	3/0
	Čačak	2	1/1	1 (50%)	1/0
	Sjenica	30	17/13	11 (36.66%)	6/5
	Raška	1	0/1	1 (100%)	0/1
	Aleksandrovac	1	1/0	/	0/0
	Kruševac	2	2/0	2 (100%)	2/0
	Novi Pazar	3	2/1	1 (33,.33%)	1/0
	Leposavić	1	0/1	/	0/0
	Blace	1	1/0	1 (100%)	1/0
	Zubin Potok	1	1//0	1 (100%)	1/0
	Tutin	2	2/0	1 (50%)	1/0
	Jagodina	1	1/0	1 (100%)	1/0
**Total**		**63**		**30 (47.6%)**	
EAST REGION	Veliko Gradište	2	1/1	1 (50%)	1/0
	Golubac	3	1/2	3 (100%)	1/2
	Petrovac na Mlavi	5	3/2	5 (100%)	3/2
	Žagubica	7	4/3	2 (28.57%)	1/1
	Despotovac	4	0/4	3 (75%)	0/3
	Paraćin	6	1/5	6 (100%)	1/5
	Zaječar	2	1/1	1 (50%)	0/1
	Boljevac	1	1/0	1 (100%)	1/0
	Ražanj	2	1/1	/	0/0
	Sokobanja	1	1/0	1 (100%)	1/0
	Knjaževac	4	3/1	3 (75%)	3/0
	Svrljig	2	1/1	2 (100%)	1/1
	Pirot	3	2/1	3 (100%)	2/1
	Bela Palanka	2	0/2	1 (50%)	0/1
**Total**		**44**		**32 (72.7%)**	
**Σ**		**107**		**62 (57.94%)**	

Abbreviation. No.—number, %—per cent, Σ—sum, M—male, F—female.

**Table 2 animals-12-03335-t002:** Alignment of partial sequences of 18S rRNA gene (539 bp) and position of variable sites.

Sequence Type	Position of the Variable Site				
	3 bp	380 bp	400 bp	522 bp	523 bp
consensus	T	A	G	C	A
S1					
S2	A				
S3	A	G	T		G
S4	A	G	T	T	G
S5	A	G	T	T	

Abbreviation. bp—base pairs, T—thymine, A—adenine, G—guanine, C—cytosine.

## Data Availability

The data presented in this study are available from the corresponding author upon request.

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
