# Peer review of "High Prevalence and Genetic Variability of Hepatozoon canis in Grey Wolf (Canis lupus L. 1758) Population in Serbia"

_animals, 2022, doi:10.3390/ani12233335_

Round 1

Reviewer 1 Report

The authors present in a very entertaining and complete way the results obtained on the presence of Hepatozoon canis in gray wolf in Serbia.

Here are my recommendations to improve the article:

First, I recommend an English revision. There are some sentences that after translation sound strange.

L12. Please change "contribution" to "contribute".

L14. "presenting finding", please reword.

L16. Please separate the words "while genetic".

L30 "sample from 30 localities in Serbia were....?" Is it an unfinished sentence? Please review.

L41. Hepatozoon is always in italics, please correct lines 41, 43, 45 and 57.

L45-46. "is very much recognized" please reword...

L46. "is insufficiently clarified in cats" please reword...

L46. Please correct "amongst" to "among".

L73, 85, 87, 99, 115, 170, 258 and 262. There is a double space.

Preparation of sample. It is not indicated what samples are collected... only the spleen? Whole spleen or just a portion? If it is a portion, what size? Hepatozoon detection is performed by PCR, conventional or real-time? If it is conventional, nothing is mentioned about gel electrophoresis. If it is real-time, nothing is indicated about the melting curve.

L115. Marogen or Macrogen?

L129. Why do you divide the wolf population in two? It is because there is some ecological difference between both groups? Please indicate the reason why you decided to divide them.

L133. Change the comma to a < symbol (p<0.05). The p-value symbol is lowercase and italic. Please correct L133, 151 and 154.

L154. Eastern and Western with lowercase, please.

L165-166. This statement is already found in L154. Please delete.

Table 1. Add below the table the meaning of No., %, and the epsilon letter.

Table 2. Add below the table the meaning of bp, T, A, G, and C.

L201. If I'm not mistaken there should be a parenthesis before MK757802 instead of the asterisk.

L217. Please change "it wasn't" to "it was not".

L222-223. Reformulate.

L224. "Pathogens"... you really only looked for one pathogen, Hepatozoon. Please correct the sentence.

L234. Please change "hapolotypes" to "haplotypes"

L242. Please change "analyzes" to "analysis".

L253-254. Please reformulate.

L258. "Wild boar is AN omnivorous species".

L281. In order to affirm that the wolf is a reservoir, you should have analyzed the transmission capacity to humans or domestic animals. The study presented only shows results on the detection of Hepatozoon in wolves, so the wolf is a host according to the results obtained. But with these data, you cannot conclude that it is a reservoir. Please remove the word reservoir from the conclusion.

Reviewer 2 Report

Dear Authors,

The idea taken up in this publication is important and interesting for readers. I believe that the manuscript, after taking into account a few bulleted comments, should be published. At the same time, I encourage the authors to continue working on this topic. These are my remarks pointed line by line:

3 - title - please add to Canis lupus L. 1758

12 - put a space between: H.canis

13, 17, 20 - correct mistakes: it should be grey , not gray

13 - 15 - three times repetition of the word the / this pathogen. Try to modify statements or write it in a different way with the help pd native speaker.

13 and 17 -  wolves, not wolfs

24 - when you write for the first time name of the species in Latin add e.g. L. 1758 or other biologist, later you can use only a shortcut - C. lupus. Also 51, 58

28 - for pathogen - I think you need add "the" or "studied"

28 - Serbia, as well as

29-30 - A total of 107 wolf spleens samples from 30 localities in Serbia were. - something is lacking here.

34 - "This study is the first that confirmed the presence..." - maybe: This study is the first confirmation of the presence

47-49 - look at remarks in 24 line

56, 59, 80, etc. - without break inside [ ], e.g. [16,17]

69 - 71 - you really don't need to repear Latin name at al, even without shortcut. Focus on it later in paper, you do this very often and this is unnecessary

73 - too big space here: distribution            occupied

74-75 - destruction of its habitats - you mean habitat loss? fragmentation?

87 - too big space here:  study      published

97 - in my opinion without 'the' here: 'about the shotted animals'. Ask a native speaker to be sure.

99 - too big spacer here: analysis     .   and 115 - to      the

Figure 1 - borders are not ideally the same, can you correct it? With Romania, Bosnia and Croatia

144-146 - this sentence I would put in chapter Material and methods, not here

Table 1 - fifth column - you mean: "No. of positive samples..."? 

Between Total and Veliko Gradiste records put a little bigger break

222 - instead of and I would put ',' and coma between 'as well as', not after

223- high, not higt

227 - Vulves vulpes - only English name is needed here. You wrote Latin name earlier. The same in lines: 229, 230

235 - [26,27]

256 - name are mentioned for the first time, add name of taxonomist

258 - why Sus scrofa again in Latin also? It was in 256 line

258 and 262 - too big spaces here: '33].       Further,' and 'H. canis.      Taking'

275 - delete : (Canis lupus)

279 - maybe instead of "According to these results" put a "According to obtained results,"

References are prepared really well, but:

314 - too big space here:   Vectors      2017

348 - lack of pages

364 - too big space here: BfN-Skripten 2015,       398, 43

398 and 399 - it should not be with such a big space, check how it looks like 

Good luck with correction
